# Influence of biosilica treatments and storage receptacles on the quality of maize (*Zea mays* L.) and common bean (*Phaseolus vulgaris* L.) seeds during long-term storage

Bertrand Zing Zing [1,2]*, Charles Rostand Mvongo Mvodo[1], Valteri Audrey Voula[1], Lin Marcellin Messi Ambassa[1], Eugene Ejolle Ehabe[1], Placide Desiré Belibi Belibi[3‡], Charles Melea Kede[2‡]

1 Directorate of Scientific Research, Institute of Agricultural Research for Development, Nkolbisson, Yaoundé, Cameroon, 2 Laboratory of Chemical and Industrial Bioprocess Engineering, National Higher Polytechnic School of Douala, University of Douala, Logbessou, Douala, Cameroon, 3 Department of Inorganic Chemistry, University of Yaoundé I, Ngoa Ekélé, Yaoundé, Cameroon

☯ These authors contributed equally to this work.
‡ These authors also contributed equally to this work.
* zingbertrand29@gmail.com

## Abstract

Common bean (*Phaseolus vulgaris* L.) is a major source of affordable, high-quality dietary protein that complements maize (*Zea mays* L.)-based diets. Traditional storage of these seeds in sacks and glass jars without treatment often leads to infestation by insect pests, such as *Acanthoscelides obtectus* and *Sitophilus zeamais,* causing significant grain losses. This study evaluated grain damage and weight loss after mixing biosilica with maize and common bean seeds, which were stored in various containers (glass jars, polyethylene, and polypropylene bags) over a six-month period, during which seed viability was also assessed. Although the initial moisture level (≤ 13%) was favourable for storage for both grain species, about half of the common bean cultivars (FEB 190 and NUV6) and two-thirds of the maize cultivars (CMS 8501 and CMS 8704) were adversely affected after prolonged storage without treatment for six months. As concerns the effects of the storage receptacle on the biosilica-treated seeds, the polyethylene bags and glass jars were more effective for common beans than for maize seeds after five months of storage. However, by the sixth month, the biosilica-containing jars were characterized by low insect perforation indices (≤ 50%) and rather high seed viability (> 80%) for both maize and common beans after six months of storage, highlighting the protective effect and storage loss reduction capacity of the biosilica. Hence, their application could help farmers improve safe grain storage over prolonged periods and eventually enhance their livelihoods and incomes.

**Data availability statement:** All relevant data are within the manuscript and its Supporting Information files.

**Funding:** Royal Society of Chemistry for the Research Fund grant (R24-9789259332) The funders had no role in study design, data collection and analysis, decision to publish, or preparation of the manuscript.

**Competing interests:** The authors have declared that no competing interests exist.

## 1. Introduction

Common bean (*Phaseolus vulgaris* L.) and maize (*Zea mays* L.) are vital dietary components, playing a crucial role in food security, income generation, and livelihoods worldwide [1]. The common bean is an important source of affordable protein that complements maize-based diets, both of which are consumed by approximately 2.5% of the global population [2]. Both grains are seasonal in production, and once harvested, farmers and processors must preserve them to ensure stable year-round consumption for their families, protect against market price fluctuations, and support planting [3]. In Cameroon, these grains are typically stored on a small scale in jute or woven polypropylene sacks, which are easy to handle, allow air circulation when properly placed, and do not require special storage spaces.

However, these sacks offer limited durability and inadequate protection against humidity and insect pests like *Sitophilus zeamais* and *Acanthoscelides obtectus* [4]. Depending on storage conditions and the type of sack used, *A. obtectus* can cause seed losses ranging from 48% to 100%, while, *S. zeamais* can damage up to 90% of unprotected maize grains in storage [5, 6]. These economically important pests bore into the kernels, feed on undamaged grains, and selectively consume nutritive components of stored common bean seeds and maize [7–9]. Furthermore, by feeding on whole or broken grains, these insect pests decrease the mass and/or volume, reduce physiological quality and seed viability, and thereby contribute significantly to food insecurity and economic strain, particularly in Sub-Saharan Africa [1, 10, 11].

To maintain the health of stored seeds, they are kept in Purdue Improved Cowpea Storage bags [12, 13], improved hermetic bags like GrainPro Super Bags, AgroZ® Plus, ZeroFly® bags, and triple-layer bags [14, 15]. Although these hermetic bags are effective, their high cost makes them less appealing to farmers in many regions of Sub-Saharan Africa. As a result, chemical pesticides such as deltamethrin and pyrimiphos-methyl are often mixed with seeds before storage in sacks. However, their residues, in the long run, pose environmental and health risks, justifying the need for the exploration of sustainable alternatives [16–18]. The effectiveness of plant volatile compounds and essential oils against these insect pests has been demonstrated, although their short shelf-life under heat, light, and moisture limits their wide use [19–21].

The insecticidal effects of biosilica on *A. obtectus* and *S. zeamais* insect pests of common bean and maize seeds have been attributed to the presence of silanol and siloxane functional groups [1], whose hydrophilic -OH group acts on both the pest and the grains. On the former, it binds to their cuticle, dehydrates them, and eventually leads to death, while on the latter, it binds and prevents evaporation of their moisture, thereby ensuring their safe storage over prolonged periods.

Knowledge on the efficiency of mixed biosilica with infested maize and common bean seeds stored in different receptacles (glass jars, polypropylene, and polyethylene bags), depicting losses under small-farm conditions over prolonged periods, is scarce. The use of rice husk-derived silica in agriculture will offer an eco-friendly and cost-efficient alternative to traditional silica sources, and as such, promote sustainable resource utilization and address waste disposal issues. Most studies applying

rice husk-derived silica to control insect pests have focused on the insects' mortality rates at different doses [22, 23], and scarcely on seeds' viability when stored in receptacles. Therefore, this study characterizes, at a laboratory scale, the evolution of the properties of some grains (maize and common beans) mixed with biosilica during prolonged storage in polypropylene, polyethylene, and glass receptacles.

## 2. Methodology

### 2.1. Collection of grain cultivars, insect rearing, and biosilica obtention

Two cultivars of maize cultivars, CMS 8501 (white) and CMS 8704 (yellow), and common beans, FEB 190 (red) and NUV6 (black), used for these experiments were obtained in June 2023 from the Crop Production Division of the Institute of Agricultural Research for Development (IRAD) in Yaoundé (Cameroon), and the Multipurpose Station of IRAD Foumbot (West Cameroon).

The insects used for bioassay tests (*A. obtectus* and *S. zeamais*) were reared, as described elsewhere [1], from February to July 2023, in the Entomology Laboratory of IRAD in Yaoundé. The adult insects were placed in 1-L glass jars containing 250 g of each grain cultivar, covered with a 2-mm mesh lid, and then stored in the dark for 60 days under controlled conditions (28 ± 2°C and 70 ± 10% relative humidity).

The procedure for obtaining the biosilica and its characterization was previously outlined by Zing et al. [1].

### 2.2. Physical properties of common beans and maize seeds

The seeds of each maize and common bean cultivar were characterized, in triplicate, for their moisture content, 100 and 1000-grain weight, bulk density, graining, and average grain weight. The moisture content was determined using a calibrated moisture meter (Dickey-John Corp, Auburn, IL 62615, USA), and the 100 and 1000-grain weights were determined by weighing the corresponding masses. Average grain weight was calculated as the ratio of the weight of 100 grains to 100, while graining was estimated as the number of seeds required to reach a weight of 100 g, and the bulk density as the ratio of the mass of each sample (in g) to its volume (in mL). All the weights were taken on a calibrated precision balance (e = 0.1 mg).

### 2.3. Storage loss estimation

About 500 g of common beans and maize seeds were weighed separately and placed in 1.2 L glass jars, into which were added 100 insects. The glass jars were then covered with muslin cloth to allow for aeration while preventing the insects from escaping, and the whole was kept at room temperature in the laboratory for six months. For each sample, the difference in grain quantity at the beginning and end of the storage period, divided by the quantity at the end, was determined at monthly intervals over the six-month period.

### 2.4. Grain damage, weight loss, and insect perforation index in storage items

This study was designed to measure the cumulative effectiveness of each treatment in controlling the pest populations following the introduction of adult insects and the creation of potentially unfavourable conditions for their larvae.

Grain samples were stored in three types of receptacles, namely: woven polypropylene (PP) bags (SPP 60 x 100, white), polyethylene (PE) bags (high-density, non-toxic, odorless, with 80% to 90% crystallinity, a softening point of 125–135°C, and an operating temperature of up to 100°C), and glass jars (GJs) procured from the Yaoundé main market. For the maize and common bean seeds placed in PE and PP bags, about 200 g each were manually and separately well mixed with 10 g of biosilica for 5 min. For those in glass jars, about 50 g of each seed type was separately put in the glass jars, and manually well mixed with 2.25 g of biosilica powders for 5 min, to ensure even distribution within the grains. Fifty and ten pairs of adult insects were introduced into both the bags and GJs, respectively, and the containers

were tightly sealed with a sheath. All the experiments were conducted in triplicate. The grain damage, weight loss, and the insect perforation index (IPI) were assessed monthly for up to 6 months, and the grain viability in the final month. All experiments were performed under controlled laboratory conditions of temperature (T = 24.1–25.6°C) and relative humidity (RH = 70.7–74.5%).

At the end of the trials, the grains were inspected for the presence of holes, and the insect-damaged ones were recorded.

Grain weight loss and the insect perforation indices were calculated using the formulas of Misele et al. [24] as well as Gever and Echezona [25].

### 2.5. Germination of treated and stored grains

The viability of treated seeds was assessed using the slightly modified method described by Berhe et al. [26]. Typically, 15 seeds of maize and common beans were randomly sampled from the GJs, PP, and PE bags and placed in sterilized Petri dishes containing a moistened cotton layer. The dishes were then randomly placed on a platform in the laboratory where the seeds were moistened daily. After 7 days, germinated grains were counted, and their germination rate was estimated as the percentage ratio of the total number of sprouted seeds to the initial number of grains.

### 2.6. Data analysis

Data were entered in an Excel spreadsheet and analyzed using XLSTAT Version 2019.2.2.59614 software. For all parameters investigated, non-linear and linear graphs were plotted using GraphPad Prism Version 10.4.2 (633) after performing a one-way ANOVA to separate significant means using Tukey's test. The post-hoc test was used to determine specific differences between cultivars, whereas an independent t-test was applied to compare the means between the two grain species at 5% probability level and 95% confidence limits. The significance of a parameter was determined at a 95% confidence threshold.

## 3. Results and discussion

### 3.1. Physical properties of common beans and maize seeds

The independent t-test results presented in Table 1 reveal significant differences (p < 0.05) between the means of graining, mass of 100-grains, mass of 1000-grains, bulk density, and moisture content of maize. This highlights substantial variation in the physical characteristics of maize seeds. In contrast, no significant differences (p > 0.05) were observed in the means of average mass, bulk densities, or moisture contents for common bean species. The higher graining obtained in common beans NUV6 suggests that more grains are required to form a homogeneous 100 g sample. The average mass and mass of 1000-grains of maize decrease with decreasing grain size, whereas in common beans NUV6, these values increase with grain size.

**Table 1. Physical properties of raw common bean (FEB 190 and NUV6) and maize (CMS 8501 and CMS 8704) grains.**

| Cultivars | Graining | Average mass (g) | Mass of 100 grains (g) | Mass of 1000 grains (g) | Bulk density (g/dm³) | Moisture content (%) |
|---|---|---|---|---|---|---|
| CMS 8501 | 387 ± 7[b] | 0.25 ± 0.07[a] | 25.9 ± 0.5[a] | 261 ± 4[a] | 763 ± 13[a] | 12.3 ± 0.4[a] |
| CMS 8704 | 441 ± 7[a] | 0.23 ± 0.04[a] | 22.7 ± 0.6[b] | 227 ± 5[b] | 714 ± 14[b] | 12.0 ± 0.6[b] |
| P (T ≤ t) | 0.0001 | 0.073 | 0.0001 | 0.001 | 0.0004 | 0.033 |
| NUV6 | 604 ± 16[a] | 0.18 ± 0.06[a] | 33.8 ± 0.6[a] | 347 ± 12[a] | 810 ± 6[a] | 12.45 ± 0.06[a] |
| FEB 190 | 563 ± 15[b] | 0.19 ± 0.01[a] | 24.3 ± 0.5[b] | 237 ± 6[b] | 784 ± 14[a] | 12.88 ± 0.07[a] |
| P (T ≤ t) | 0.002 | 0.423 | 0.001 | 0.002 | 0.112 | 0.298 |

Means followed by the same letters in each column are not significantly different according to the independent t-test at P < 0.05.

A significant difference (p < 0.05) was also found in the 100-grain weight between white maize CMS 8501 (25.9±0.5g) and yellow maize CMS 8704 (22.7±0.6g), with white maize kernels weighing approximately 1.14-times more than yellow maize kernels. However, no significant difference (p>0.05) was observed in the 100-grain weight between common bean cultivars FEB 190 (18.3±0.5g) and NUV6 (17.8±0.6g). According to the size classification of Kläsener et al. [27], all evaluated common bean samples have medium-sized grains (< 40 g/100 grains). Another significant difference (p < 0.05) was observed in the 1000-grain weight of maize seeds, with CMS 8501 having the highest value. No such difference was found among common bean seeds. Moisture contents of all kernels were below 13% and did not differ significantly (p>0.05) between maize cultivars (CMS 8704 and CMS 8501) or between common bean cultivars (FEB 190 and NUV6). The moisture contents of all the seeds were relatively low (below 13%), indicating their suitability for storage, with limited development and proliferation of pathogenic grain fungi [28].

Bulk densities varied significantly (p < 0.05) between maize cultivars (CMS 8501: 763±13 g/dm³; CMS 8704: 714±14 g/dm³), but no significant (p>0.05) variation was observed among common bean cultivars (FEB 190: 784±14 g/dm³; NUV6: 810±6 g/dm³). This parameter is used to classify grains for trading and processing into food products [29]. In this study, the moisture content of the CMS 8501 maize cultivar increased with higher bulk density, but it slightly decreased with the bulk density of the CMS 8704 cultivar. Conversely, the bulk density of both common bean cultivars decreased as their moisture contents decreased. The determination of these parameters is essential during trading to ensure fair pricing for both farmers and seed companies.

### 3.2. Storage losses of common beans and maize seeds

Fig 1 shows an increase in losses of common beans and maize grains to which insects were added at trial set-up in the glass jars for 6 months.

The post-hoc analysis of data plotted in Fig 1 shows no significant differences (F=0.70; DF=3; p=0.56) in losses between maize and among common bean cultivars, indicating that for the maize and common bean species, mean losses simply increased with storage duration. Indeed, the maize seeds suffered more than 50% loss after 6 months of storage (Fig 1A), with the greatest loss being 85% for CMS 8704 and 65% for CMS 8501 seeds initially seeded with the same number of *S. zeamais*. The higher loss of CMS 8704 is likely related to its higher carbohydrate content (75% as

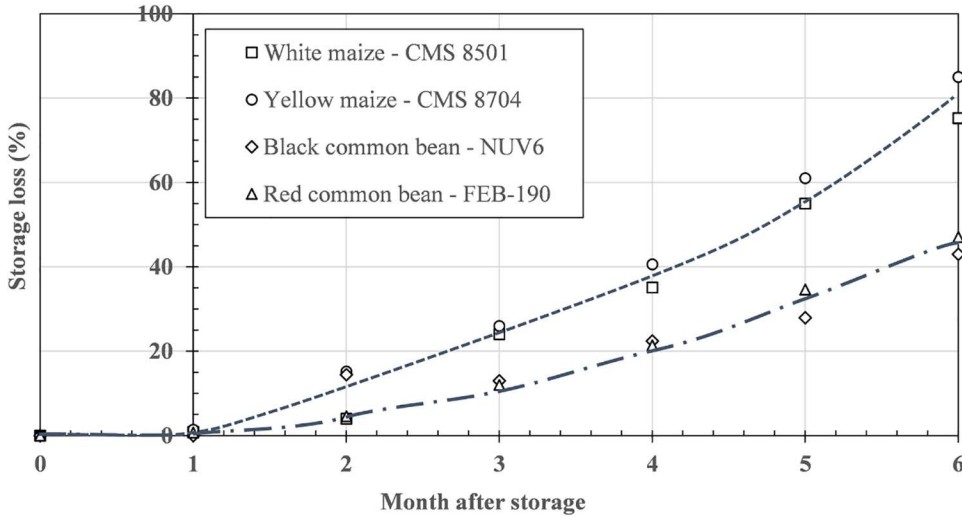

**Fig 1. Evolution of storage losses for (A) maize and (B) common bean cultivars during 6 months in the laboratory.**

compared to 65% for CMS 8501) [30], in the form of starch, sugar, cellulose, and non-cellulosic polysaccharides in the pericarp. Another factor that lowers grain losses of CMS 8501 compared to CMS 8704 could be the white color of the grains. In fact, Cortese et al. [31] have already demonstrated that the brightness of the white color in maize grains may have a repellent effect on adults of *S. zeamais*, known as an antixenosis effect. In contrast, common bean seeds showed less than 50% loss over 6 months of storage (Fig 1B), probably due to their relatively higher resistance to perforation by *A. obtectus*, which makes them more difficult to bore. Further research is required to confirm this. The increased storage losses in maize grains may be further attributed to the longer lifespan and prolonged activity of *S. zeamais* adults (3–6 months or more) [32] as well as their robust mandibles that attack the compact grains [33]. The results obtained here showed that storage losses were higher for maize seeds than for common beans, as from the second month of storage.

### 3.3. Effect of biosilica and receptacles on grain damage during storage

The evolution of damaged grains in stored items treated with biosilica powder and controls (without biosilica) is shown in Fig 2.

The evolution of damaged grains treated with rice husk derived biosilica in different storage receptacles, indicates that maize seeds treated and stored in glass jars (Fig 2A) experienced an exponential increase in damage, from the onset to attain the maximum (total damage) after six months of storage, irrespective of the cultivar tested, irrespective of the type of receptacle in which the grains were stored. The damages caused by *S. zeamais* on maize grains are consistent with those reported by Quellhorst et al. [34]. Results obtained on the common bean varieties (Fig 2B) were much different, as the increase in damage, though exponential, varied with the cultivar tested. Here, the red variety (FEB 190) experienced a maximum damage of about 30–35% after 6 months of storage, whereas the black variety (NUV6) attained a maximum damage of just about 10–13% after storage for the same period of time. The storage receptacle seemed to contribute to the proportion of damaged grains, as those stored in polyethylene bags were the most affected, followed by those in polypropylene bags, while the least damage was recorded for grains in glass jars. However, the proportion of damage on these black common beans was much lower than that reported by Kuyu et al. [35] when evaluating different grain storage technologies against insect pests.

The differences in the results obtained on the common bean may be attributed to temperature fluctuations, aeration, and oxygen diffusion in the various receptacles. The glass jars will minimize temperature fluctuations between the interior and exterior of the receptacles, while the limited oxygen supply will eventually reduce the insects' survival and movements as well as increase the insecticidal activity of biosilica [36].

### 3.4. Effect of biosilica and receptacles on weight loss during storage

The mean percentage weight losses of common beans and maize seeds treated with biosilica powder over the storage period are shown in Fig 3.

Fig 3 presents the effects of different storage receptacles on the evolution during storage of weight losses of biosilica-treated maize (Fig 3A) and common bean grains (Fig 3B), a phenomenon often associated with endosperm consumption and breakdown of carbohydrate and protein reserves in plants [37].

For the treated common beans, weight losses rose to a maximum of about 9% for those stored in polyethylene bags, about 6% for those in polypropylene bags, and just 1% for those in glass jars. This was not the case for the treated maize grains, which recorded weight losses of less than 5% throughout the storage period, irrespective of the storage receptacle. These results could indicate respiration was higher in common bean than in maize grains, resulting in weight loss of only 10% for both red and black common bean seeds. Elsewhere, Muhammad et al. [38] found that red and black Bambara groundnut (*Vigna subterranea*) landraces lost 57% and 61% of their weight, respectively, after being attacked by *Callosobruchus maculatus*. However, the weight losses in this study are much lower than the 20–80% reported by Taddese et al. [39] for maize stored traditionally for the same period of time.

 

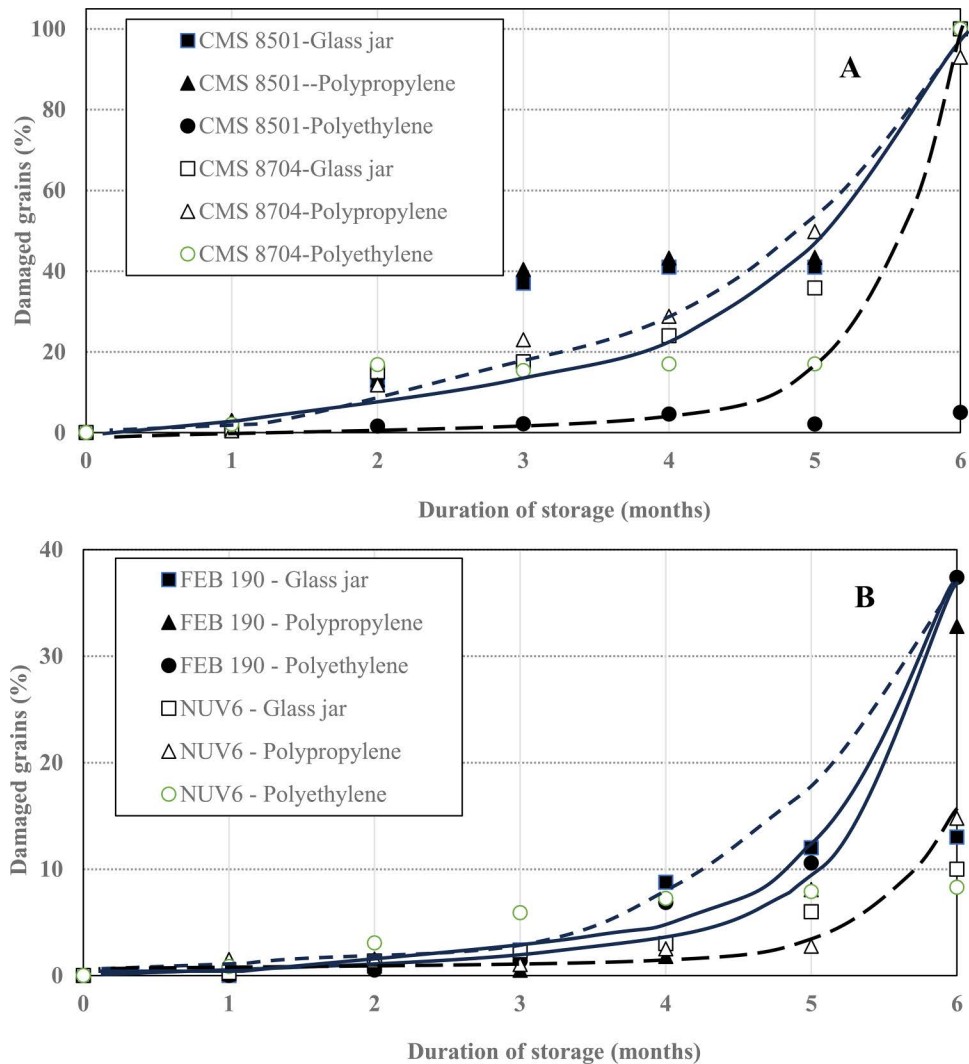

**Fig 2. Evolution of damaged grains for (A) treated maize and (B) treated common bean seeds.**

### 3.5. Variation of insect perforation index on treated common beans and maize seeds

The mean perforation indices of stored common beans and maize seeds treated throughout the storage months are displayed in Fig 4.

The evolution of mean perforation indices during storage of treated common beans and maize grains was similar for both the maize (Fig 4A) and common bean cultivars (Fig 4B), although more distinct for the former. Except for maize grains stored in polyethylene bags, the insect perforation indices for maize seemed to increase exponentially to a maximum at 40–50%, after about two months of storage, and then remain stable thereafter, irrespective of the storage receptacle used. The trends for the common bean cultivars were similar but characterized by a larger amplitude (25–50%) that was attained after storage for 1–2 months as well, and then remained rather constant thereafter. Considering that an insect perforation index of less than 50% indicates a positive protectant effect [40], the results presented here thereby indicate that the biosilica had a more protective effect against the respective pests than the storage receptacles used. The

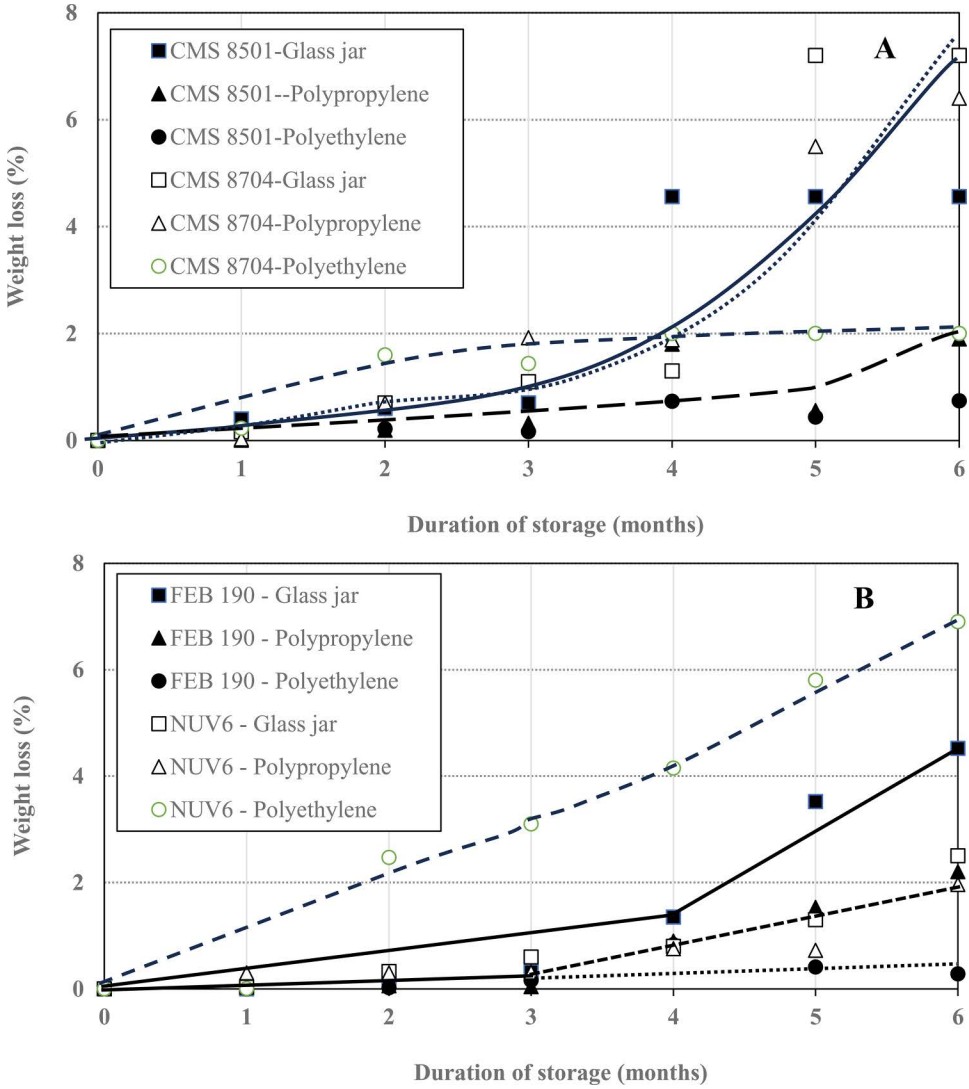

**Fig 3. Evolution of weight loss for (A) treated maize and (B) treated common bean seeds.**

overall reduction in perforation indices across the three receptacles for all grains suggests an ovicidal or even larvicidal effect of biosilica, which prevents insect molting, development, and consequently, emergence.

### 3.6. Viability of common beans and maize seeds stored in different receptacles

The viability of seeds that had been treated with biosilica powder and subjected to prolonged storage in the different containers was evaluated after 7 days of placement under conditions that favour germination.

The results obtained, presented in Table 2, showed significant differences that could be attributed to the storage receptacles as well as the plant cultivars. Irrespective of the plant species and cultivar tested, seed viability was systematically higher for seeds stored in polythene bags, followed by those stored in polypropylene bags, whereas the lowest values were obtained from those stored in glass jars. Among the maize cultivars, CMS 8501 retained higher viability than CMS 8704, irrespective of the storage recipient used. The phenomenon was the same for the common beans, where FEB 190 also retained a higher viability after prolonged storage over the NUV6, irrespective of the storage recipient.

                                                              

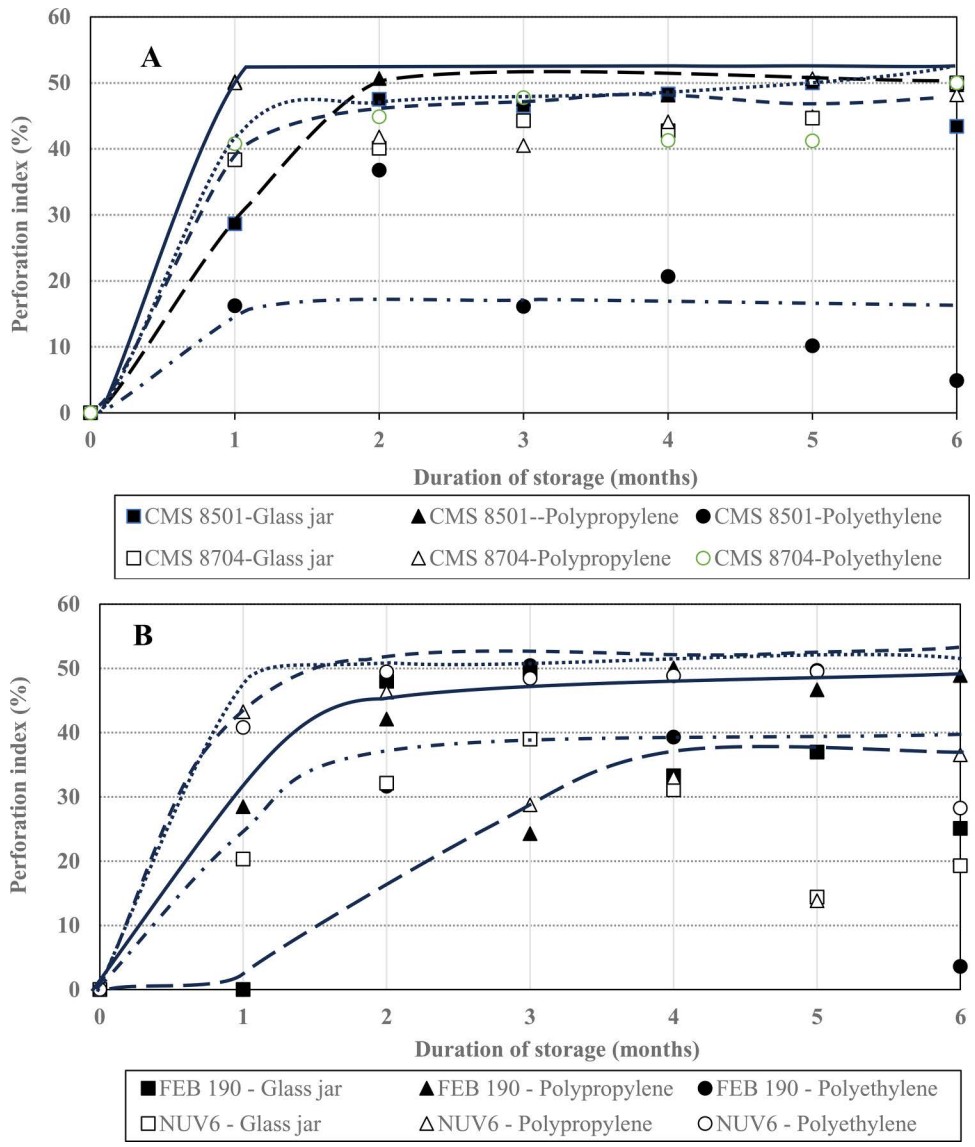

**Fig 4. Evolution of perforation index on (A) maize (CMS 8704 and CMS 8501) and (B) common beans (NUV6 and FEB 190) grains.**

The differences in viability rates for seeds stored in the different storage receptacles indicate clearly that the viability of the seeds during prolonged storage is closely associated with the porosity and permeability of the receptacles, as conventional solid glass is essentially of a higher porosity than polypropylene. At the same time, both have a higher intrinsic viscosity than polyethylene, which allows for less exchanges with the environment [41]. Indeed, in aerated environments, the insect pests have more oxygen to survive on, and the respiration of the seeds consumes their endosperm, and eventually reduces their germinative capacity [42].

## Conclusion

Studies conducted on the effects of biosilica treatment on the damage and viability of maize and common bean seeds after prolonged storage in three different storage receptacles (glass jars, polyethylene bags, and polypropylene bags). The seed

**Table 2. Effect of storage receptacles on the germination rate, in percentage, of maize and common bean grains.**

| Grain species | Cultivars | Glass jars | Polypropylene bags | Polyethylene bags |
|---|---|---|---|---|
| Maize | CMS 8501 | 56 ± 9[a] | 67 ± 11[a] | 80 ± 7[a] |
| | CMS 8704 | 40 ± 6[b] | 58 ± 4[a,b] | 62 ± 4[b] |
| | P (T ≤ t) | 0.037 | 0.271 | 0.041 |
| Common beans | NUV6 | 31 ± 4[b] | 51 ± 4[b] | 62 ± 4[b] |
| | FEB 190 | 73 ± 8[a] | 87 ± 6[a] | 91 ± 4[a] |
| | P (T ≤ t) | 0.011 | 0.015 | 0.006 |

Means followed by the same letters in each column are not significantly different according to the independent t-test at p < 0.05.

losses were more associated with the storage duration than with the type of receptacles, irrespective of the plant species or cultivars, highlighting the effects of the prolonged activity of adult insect pests during storage. Grain weight loss during storage, often associated with endosperm consumption and breakdown during the plants' respiration, was different for the two plant species and varied with the storage receptacle. For the common beans, weight losses were highest in polyethylene bags, intermediate in polypropylene bags, and lowest in glass jars. For the maize grains, the losses were intermediate and the same, irrespective of the storage receptacle used, indicating thereby that respiration was higher in common bean than in maize. The evolution of perforation indices during storage was similar for common bean and maize grains, although more distinct for maize. Increasing exponentially after about two months to a maximum at 40–50% for maize, and with a larger amplitude (25–50%) for common beans after the same duration of storage, irrespective of the storage receptacle used. Thereafter, the perforation indices remained constant. Furthermore, the common bean seeds were slightly more viable after prolonged storage in different containers than the maize seeds. Further tests must be conducted on these stored receptacles when loaded with commercial silica, and the results compared with those obtained using our biosilica.

## Supporting information

**S1 File. The core datasets of weight loss and damaged grains.**
(XLSX)

**S2 Table. Post-hoc analysis layout for maize and common bean cultivars after performing an independent ANOVA.**
(DOCX)

**S3 Table. Tukey's honest significant difference (HSD) post-hoc analysis layout.**
(DOCX)

**S4 Table. Means of common beans and maize damaged after treatment with biosilica and stored for 06 months.**
(DOCX)

**S5 Table. Means weight losses of common beans and maize seeds treated with biosilica after 06 months of storage.**
(DOCX)

**S6 Table. Means of insect perforation index on treated common beans and maize seeds after 06 months.**
(DOCX)

**S7 Table. Means of viability on treated common beans and maize seeds stored in different receptacles after 06 months.**
(DOCX)

## Acknowledgments

Special thanks to the management of the Institute of Agricultural Research for Development (IRAD), Cameroon, for their support.

## Author contributions

**Conceptualization:** Bertrand Zing Zing.

**Data curation:** Bertrand Zing Zing, Eugene Ejolle Ehabe.

**Formal analysis:** Bertrand Zing Zing.

**Funding acquisition:** Bertrand Zing Zing.

**Investigation:** Bertrand Zing Zing.

**Methodology:** Bertrand Zing Zing.

**Resources:** Bertrand Zing Zing.

**Software:** Bertrand Zing Zing, Eugene Ejolle Ehabe.

**Supervision:** Eugene Ejolle Ehabe, Placide Désiré Belibi Belibi, Charles Melea Kede.

**Validation:** Charles Rostand Mvongo Mvodo, Lin Marcellin Messi Ambassa, Eugene Ejolle Ehabe, Placide Désiré Belibi Belibi, Charles Melea Kede.

**Visualization:** Bertrand Zing Zing, Eugene Ejolle Ehabe.

**Writing – original draft:** Bertrand Zing Zing.

**Writing – review & editing:** Bertrand Zing Zing, Charles Rostand Mvongo Mvodo, Lin Marcellin Messi Ambassa, Eugene Ejolle Ehabe, Placide Désiré Belibi Belibi, Charles Melea Kede, Valteri Audrey Voula.

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
