## [Decision Letter · Decision Letter 0]

16 Sep 2025

Dear Dr. Bertrand,

Thank you for submitting your manuscript to PLOS ONE. After careful consideration, we feel that it has merit but does not fully meet PLOS ONE’s publication criteria as it currently stands. Therefore, we invite you to submit a revised version of the manuscript that addresses the points raised during the review process.

We look forward to receiving your revised manuscript.

Kind regards,

Yêyinou Laura Estelle Loko

Academic Editor

PLOS ONE

**Journal Requirements:**

1. When submitting your revision, we need you to address these additional requirements. Please ensure that your manuscript meets PLOS ONE's style requirements, including those for file naming. The PLOS ONE style templates can be found at https://journals.plos.org/plosone/s/file?id=wjVg/PLOSOne_formatting_sample_main_body.pdf and https://journals.plos.org/plosone/s/file?id=ba62/PLOSOne_formatting_sample_title_authors_affiliations.pdf 2. Thank you for stating the following financial disclosure: Royal Society of Chemistry for the Research Fund grant (R24-9789259332)   Please state what role the funders took in the study.  If the funders had no role, please state: "The funders had no role in study design, data collection and analysis, decision to publish, or preparation of the manuscript." If this statement is not correct you must amend it as needed. Please include this amended Role of Funder statement in your cover letter; we will change the online submission form on your behalf. 3. Please amend the manuscript submission data (via Edit Submission) to include author Dr. Valteri Audrey Voula. 4. Please include captions for your Supporting Information files at the end of your manuscript, and update any in-text citations to match accordingly. Please see our Supporting Information guidelines for more information: http://journals.plos.org/plosone/s/supporting-information. 5. If the reviewer comments include a recommendation to cite specific previously published works, please review and evaluate these publications to determine whether they are relevant and should be cited. There is no requirement to cite these works unless the editor has indicated otherwise. 

**Additional Editor Comments:**

**Reviewer #1:** In overall, i think this manuscript didn't meet high impact journal such as Plos One. It is too simple experiment. and less novelty value. May be, the author should add more treatments which can compare biosilica. Without biosilica as control is too simple analysis. Detail data such as oxygen content, fecundity inset etc could also be good for this study. Anyway, this paper still good for publication and probably just minor correction for Q2 paper index. The author can refer my comments in main manuscript.

**Reviewer #2:** This study is well designed, has a new innovation. It was successfully laboratory conducted, and was written in good language.

Comments:

Introduction

Line 77 “ to our knowledge……” transfer this sentence to the end of introduction (before: therefore) and include it to clarify the objective of the study

Give a more brief account on Biosilica, its source, previous studies etc…

Results and Discussion

Line 313: Please write this sentence “Our findings show have a better outcome than the work of Muhammad et al. [39] which established 61% and 57% damage on red and black landraces of Bambara groundnut (Vigna subterranean) seeds respectively following attack by Callosobruchus maculatus which translated to weight loss suffered by the affected seeds.” In another manner; for example: “The work of Muhammed et al…….. However, our findings show that…..).

Line 370: “Biosilica used in this study did not have negative affect (on) maize seed germination.” Add (on)

Line 372: “However, no germination was recorded for either common bean seeds or untreated maize grain (negative control) stored for 6 months in GJs, PP and PE bags”. Do you mean: either untreated common beans or maize grains?

Thanks

Reviewers' comments:

**Comments to the Author**

1. Is the manuscript technically sound, and do the data support the conclusions?

Reviewer #1: Partly

Reviewer #2: Yes

2. Has the statistical analysis been performed appropriately and rigorously?

Reviewer #1: Yes

Reviewer #2: Yes

3. Have the authors made all data underlying the findings in their manuscript fully available?

Reviewer #1: No

Reviewer #2: Yes

4. Is the manuscript presented in an intelligible fashion and written in standard English?

Reviewer #1: Yes

Reviewer #2: Yes

**Reviewer #1:**  In overall, i think this manuscript didn't meet high impact journal such as Plos One. It is too simple experiment. and less novelty value. May be, the author should add more treatments which can compare biosilica. Without biosilica as control is too simple analysis. Detail data such as oxygen content, fecundity inset etc could also be good for this study.

Anyway, this paper still good for publication and probably just minor correction for Q2 paper index. The author can refer my comments in main manuscript.

**Reviewer #2:** This study is well designed, has a new innovation. It was successfully laboratory conducted, and was written in good language.

Minor corrections

Comments:

Introduction

Line 77 “ to our knowledge……” transfer this sentence to the end of introduction (before: therefore) and include it to clarify the objective of the study

Give a more brief account on Biosilica, its source, previous studies etc…

Results and Discussion

Line 313: Please write this sentence “Our findings show have a better outcome than the work of Muhammad et al. [39] which established 61% and 57% damage on red and black landraces of Bambara groundnut (Vigna subterranean) seeds respectively following attack by Callosobruchus maculatus which translated to weight loss suffered by the affected seeds.” In another manner; for example: “The work of Muhammed et al…….. However, our findings show that…..).

Line 370: “Biosilica used in this study did not have negative affect (on) maize seed germination.” Add (on)

Line 372: “However, no germination was recorded for either common bean seeds or untreated maize grain (negative control) stored for 6 months in GJs, PP and PE bags”. Do you mean: either untreated common beans or maize grains?

Thanks

**Do you want your identity to be public for this peer review?** For information about this choice, including consent withdrawal, please see our Privacy Policy

Reviewer #1: No

Reviewer #2: **Yes:** Marah Mohammad Hassan Abd El-Bar

---

## [Author Response · Author response to Decision Letter 1]

9 Oct 2025

Effect of stored material using biosilica on grain losses for the eco-conservation of maize (Zea mays L.) and common bean (Phaseolus vulgaris L.) seeds

Journal Requirements:

Response: We appreciate your remark dear Editor. The manuscript body has been formatted as PLOS ONE’s style requirements, including those for file naming.

Royal Society of Chemistry for the Research Fund grant (R24-9789259332)

If this statement is not correct, you must amend it as needed.

Response: Thank you for your remark, dear Editor. The role of the funder has been mentioned as follows in the cover letter: “Royal Society of Chemistry’s Research Fund grant (R24-9789259332) played a crucial role in the collection and analysis of data”.

3. Please amend the manuscript submission data (via Edit Submission) to include author Dr. Valteri Audrey Voula.

Response: Many thanks for your remark, dear Editor. The name and surname of Dr. Valteri Audrey Voula have been uploaded to the system.

Response: We are so grateful for your remark, dear Editor.

Response: Thank you for your remark dear Editor. The comments including recommendation of the Reviewer 2 have been taken in consideration in the manuscript revised”.

Reviewer #1: In overall, I think this manuscript didn’t meet high impact journal such as Plos One. It is too simple experiment. and less novelty value. May be, the author should add more treatments which can compare biosilica. Without biosilica as control is too simple analysis. Detail data such as oxygen content, fecundity inset etc. could also be good for this study. Anyway, this paper still good for publication and probably just minor correction for Q2 paper index. The author can refer my comments in main manuscript.

Line 1 and 2: The title should be

Response: Thank you for your remark dear Reviewer 1. The effect of stored material using biosilica on grain losses for eco-conservation of maize (Zea mays L.) and common bean (Phaseolus vulgaris L.) seeds

Line 78: Please extend the information on biosilica used for store seed treatment. Because it is the main focus of this study. May be a type of biosilica and rate. Or pros and cons of using biosilica as a store seed material.

Response: Many thanks for your remark, dear Reviewer 1. A brief account of Biosilica has been inserted.

Line 96: Be consistent with the order because the results start with maize followed by common bean

Response: We are grateful for your remark, dear Reviewer 1. This was done accordingly.

Line 185 and 186: Repeat information for 100-grain weight

Response: Thank you for your remark, dear Reviewer 1. This was a typing error that is now corrected as 1000-grain weight in the manuscript.

Line 189: No statistical analysis has been conducted for comparing maize and common bean seeds.

Response: We appreciated your comment, dear Reviewer 1. As seeds type have different characteristics and are not infested by the same insect pests, an independent ANOVA followed by a post-hoc test was performed for each crop species. The post-hoc analysis of table 1 (Table 1a and 1b) reveals high significant difference between physical parameters of maize (p = 3.3497E-7) and among common bean seeds (p = 1.6425E-06).

Table 1: Layout of post-hoc analysis for maize and common bean cultivars.

(a) Maize cultivars (CMS 8501: white and CMS 8704: yellow)

Source of variations Sum of Squares DL Mean of squares F-Value P>F F-crit.

Between groups 869426.982 5 173885.396 321.844581 3.3497E-7 4.38737419

Within groups 3241.6652 6 540.277533

Total 872668.647 11

(b) Common bean cultivars (FEB-190: red and NUV6: black)

Source of variations Sum of Squares DL Mean of squares F-Value P>F F-crit.

Between groups 1144176.72 5 228835.345 188.763458 1.6425E-06 4.38737419

Within groups 7273.7175 4 1212.28625

Total 1151450.44 11

Lines 192-194: About the significant difference in bulk densities for maize and for common bean seeds.

Response: Many thanks for your remark dear Reviewer 1. The sentence was rewritten as bulk densities varied significantly (p ˂ 0.05) between maize cultivars (CMS 8501: 763 ± 13 g/dm3; CMS 8704: 714 ± 14 g/dm3), but no significant (p > 0.05) variation was observed among common bean cultivars (FEB-190: 784 ± 14 g/dm3; NUV6: 810 ± 6 g/dm3).

Lines 205-206: Have more information about the significance of storage losses between maize and within common bean seeds

Response: Thank you for this comment, dear Reviewer 1. The Tukey test was performed on all the data related to storage losses and gave the same outputs as the Duncan test. We then performed a post-hoc test to determine the specific differences between cultivars. As shown in Table 2, there was no significant difference (p = 0.563) between maize and common bean seeds, or within them. This suggests that, for each species of maize and common bean, losses increase with storage duration.

Table 3: Layout of post-hoc analysis.

Source of variations Sum of Squares DL Mean of squares F-Value P>F F-crit

Between groups 1258.91838 3 419.639461 0.696058533 0.563530999 3.00878657

Within groups 14469.1094 24 602.879558

Total 15728.0278 27

Lines 217-219: The authors should include the nutritional content data or quoted data specific for maize to support this claim.

Response: Thank you for this remark dear Reviewer 1. The quoted nutritional contents for yellow and white maize seeds were appropriately quoted to support our claim.

Line 224: Need quoted paper or previous study to support this claim or just mention further study need to be done.

Response: We appreciate this remark dear Reviewer 1. We mention the need of further studied to better clarify this claim.

Line 228: Based on insect preference testing (yellow) or seed resistant characterization such as soft skin grain.

Response: We are grateful for this remark dear Reviewer 1.

Line 245: I’m facing difficulty to understand how the author presents this result. Unspecific and scattered presentation of results. For me better focus on Maize first and then bean. So, we know which is the best material pun storing maize. Very simple experiment. For me, the best explanation for maize at 5 months of store. PE is the best material for both treated and untreated.

However, for bean store experiment, the damaged pattern was not clear. May be data variation quite high. The author should explain why it wat happen. Overall, PE material was good as well as GJ. But I don’t know why FEB190 show the high damaged for untreated.

Response: Thank you for this remark dear Reviewer 1. For bean store experiment, A. obtectus insects can hibernate and multiply inside common bean seeds for some time. The insects are often trapped inside the seeds, which increases the damage of common bean seeds. It is why FEB190 shows the high damage for untreated grains.

Line 294: I think variation or CV value of data were quite high. That the reason why distribution of were not in good pattern. Difficult for to understand. My suggestion, they should convert this data first and then do analysis again. The Mann-Whitney analysis as example.

Response: Thank you for this remark dear Reviewer 1. The Mann-Whitney analysis was done with data transformed (Table 3) and data without transformation (Table 4) for the weight loss.

Table 3. Results of the Mann-Whitney test for the comparison of weight loss with data transformed

Cultivars Store materials Mann-Whitney test p value Cultivars Store materials Mann-Whitney test p value

Maize GJs 0.588 Common beans GJs 0.032*

GJs Check 0.996 GJs Check 0.760

PP 0.755 PP 0.402

PP Check 0.980 PP Check 0.862

PE 0.299 PE 0.261

PE Check 0951 PE Check 0.830

*There was a significant difference between the weight loss results for the compared assays in the control.

Table 4. Results of the Mann-Whitney test for the comparison of weight loss data non-transformed

Cultivars Store materials Mann-Whitney test p value Cultivars Store materials Mann-Whitney test p value

Maize GJs 0.705 Common beans GJs 0.241

GJs Check 0.767 GJs Check 0.110

PP 0.026* PP 0.367

PP Check 0.395 PP Check 0.516

PE 0.0001** PE ˂0.0001***

PE Check 0.335 PE Check 0.159

*There was a significant difference between the weight loss results for the compared assays in the control.

**There was a high significant difference between the weight loss results for the compared assays in the control.

****There was a very high significant difference between the weight loss results for the compared assays in the control.

Line 303: Should have correlation table between weigh loses and insect damaged.

Response: Thank you for this remark dear Reviewer 1. The Pearson’s correlation table between weigh loses and insect damaged of maize (Table 5; page 7) and common bean seeds (table 6; page 7) was done

Lines 347-352: In my opinion, it is better to collect percentage germination for each month store rather than only after 6 months. For me, days after sprouting is not important in this study. Percentage germination after 7 days could be enough

Response: Thank you for this remark, dear Reviewer 1. For our upcoming studied, data on percentage germination will be collected each month and estimated after 07 days.

Line 378: No need to explain again, caused is already in methodology. May be material could be enough

Response: Many thanks for your remark, dear Reviewer 1.

Lines 386-387: The authors should conduct this experiment to confirms this claim.

Response: We appreciate your remark, dear Reviewer 1.

Reviewer #2: This study is well designed, has a new innovation. It was successfully laboratory conducted, and was written in good language.

Comments:

Introduction

Line 77 “ to our knowledge……” transfer this sentence to the end of introduction (before: therefore) and include it to clarify the objective of the study

Give a more brief account on Biosilica, its source, previous studies etc…

Response: We are so grateful for your remark, dear Reviewer 2. Your comments including recommendation have been taken in consideration in the manuscript revised”.

Results and Discussion

Line 313: Please write this sentence “Our findings show have a better outcome than the work of Muhammad et al. [39], which established 61% and 57% damage on red and black landraces of Bambara groundnut (Vigna subterranean) seeds, respectively, following attack by Callosobruchus maculatus, which translated to weight loss suffered by the affected seeds.” In another manner; for example: “The work of Muhammed et al…….. However, our findings show that…..).

Response: Thank you for your remark, dear Reviewer 2. The whole sentence was rephrased”.

Line 372: “However, no germination was recorded for either common bean seeds or untreated maize grain (negative control) stored for 6 months in GJs, PP and PE bags”. Do you mean: either untreated common beans or maize grains? Thanks

Response: We appreciated your remark, dear Reviewer 2. Yes. “However, no germination was recorded for either untreated common bean or maize seeds (negative control) stored for 6 months in GJs, PP and PE bags”.

Table 5. Pearson’s correlation between weigh loses and insect damaged of maize seeds

Where, DGn denoted Damaged grains at month n (n=1 to 6) and WLn the Weight loss at month n (n=1 to 6).

Table 6. Pearson’s correlation between weigh loses and insect damaged of common bean seeds

---

## [Decision Letter · Decision Letter 1]

25 Dec 2025

Dear Dr. Bertrand,

Thank you for submitting your manuscript to PLOS ONE. After careful consideration, we feel that it has merit but does not fully meet PLOS ONE’s publication criteria as it currently stands. Therefore, we invite you to submit a revised version of the manuscript that addresses the points raised during the review process.

We look forward to receiving your revised manuscript.

Kind regards,

Yêyinou Laura Estelle Loko

Academic Editor

PLOS One

Journal Requirements:

Additional Editor Comments :

Research evaluation

Efficiency of Biosilica on Grain Losses for Eco-conservation of Maize (Zea mays L.) and Common Bean (Phaseolus vulgaris L.) Seeds

Executive Summary

This manuscript presents an applied and valuable study exploring the use of a rice husk ash derivative, biosilica, for protecting stored grains from pests. The research addresses a problem with significant economic and nutritional implications, especially for small-scale farmers. The manuscript has a solid foundation for publication but requires the resolution of some fundamental gaps to enhance its clarity and compliance with publishing standards.

Overall Assessment

Initial Decision: Accept after Major Revisions.

Suitability for the Journal: The research aligns well with the scope of PLOS ONE, presenting experimental work in the field of applied agricultural sciences.

Significance of Contribution: High, as it proposes a practical, potential solution to a global post-harvest problem.

Nature of Feedback: Some points are critical for acceptance, while others aim to improve the quality of analysis and presentation.

Main Strengths of the Research

1. Addresses a Relevant Problem: Directly targets the challenge of post-harvest grain loss, which impacts livelihoods and food security.

2. Proposes a Sustainable Alternative: Offers a potential eco-friendly solution by utilizing an agricultural by-product (rice husk) as an alternative to chemical pesticides.

3. Appropriate Experimental Design: Effectively tests the interaction of key variables (container type, grain type, treatment effect) over a realistic storage period.

4. Clear and Actionable Results: Leads to practical recommendations that farmers could implement, including an estimate of the protection period.

Primary Weaknesses and Development Suggestions

1. Critical Issue with Data Availability Statement:

Description: A contradiction exists between the submission system response (stating data is in the manuscript) and the dedicated data availability statement in the manuscript (stating it is available upon request). This conflicts with PLOS ONE's publication policy.

Proposed Solution: The authors must upload the core dataset (e.g., counts of damaged grains, weights for each replicate) as Supporting Information files (e.g., an Excel file). The Data Availability Statement must then be updated to accurately reflect this, ensuring consistency across all sections.

2. Methodological Clarity Regarding Treatment Impact:

Description: Adult pests were introduced only at the start, yet the discussion posits the material kills pests. Clarification is needed on how the results are interpreted concerning the pest's full life cycle (including eggs and larvae inside grains).

Proposed Solution: Adding clarification in the methodology or discussion stating that the design measures the cumulative ability of the treatment to suppress pest population development within the semi-closed environment, through its effect on adults and possibly by creating unfavorable conditions for larvae.

3. Improving Results Presentation (Figures):

Description: Many figures (e.g., 2, 3, 4, 5) are overly complex with numerous plot lines, making it difficult for the reader to follow key outcomes.

Proposed Solution: It is suggested to split complex figures into simpler sub-figures (e.g., separating maize from beans, or presenting storage materials separately). Bar charts could help present comparisons more clearly at specific time points.

4. Deepening the Discussion:

Description: The discussion could be enriched by a preliminary mention of the method's potential economic feasibility compared to alternatives, and its expected performance under different humidity or temperature conditions than those in the lab.

Proposed Solution: Adding a short paragraph in the concluding section or discussion touching on these points would enhance the applied value of the research.

Editorial Notes

Language: Some minor typographical and grammatical errors require correction (e.g., "obectus" should be corrected to "obtectus").

References: The formatting of some references is incomplete or inconsistent (e.g., references 1, 22, 39). They should be reviewed and standardized according to the journal's style guide.

Clarity: Ensure all abbreviations (GJ, PE, PP) are defined upon first use, and that all figure legends are complete and clear.

Final Recommendation

I recommend accepting the manuscript pending Major Revisions. This research has clear applied importance. Resolving the issues related to data policy and improving methodological clarity and presentation will make it a strong, publishable contribution to the literature on sustainable post-harvest management

Reviewers' comments:

Reviewer's Responses to Questions

**Comments to the Author**

Reviewer #3: All comments have been addressed

Reviewer #4: (No Response)

2. Is the manuscript technically sound, and do the data support the conclusions?

Reviewer #3: Yes

Reviewer #4: Yes

3. Has the statistical analysis been performed appropriately and rigorously?

Reviewer #3: Yes

Reviewer #4: Yes

4. Have the authors made all data underlying the findings in their manuscript fully available?

Reviewer #3: Yes

Reviewer #4: No

5. Is the manuscript presented in an intelligible fashion and written in standard English?

Reviewer #3: Yes

Reviewer #4: Yes

Reviewer #3: Well versed manuscript following reviewers suggestions. Authors have improved and prepared it in the required and presentable way.

Reviewer #4: I recommend accepting the manuscript pending Major Revisions. This research has clear applied importance. Resolving the issues related to data policy and improving methodological clarity and presentation will make it a strong, publishable contribution to the literature on sustainable post-harvest management.

**Do you want your identity to be public for this peer review?** For information about this choice, including consent withdrawal, please see our Privacy Policy

Reviewer #3: No

Reviewer #4: No

---

## [Author Response · Author response to Decision Letter 2]

28 Jan 2026

1. Critical Issue with Data Availability Statement:

Description: A contradiction exists between the submission system response (stating data is in the manuscript) and the dedicated data availability statement in the manuscript (stating it is available upon request). This conflicts with PLOS ONE’s publication policy.

Proposed Solution: The authors must upload the core dataset (e.g., counts of damaged grains, weights for each replicate) as Supporting Information files (e.g., an Excel file). The Data Availability Statement must then be updated to accurately reflect this, ensuring consistency across all sections.

Response: We appreciate your remark, dear Editor. The core dataset, including counts of damaged grains, weights for each replicate, has been uploaded to the system as Supporting Information files. The Data Availability Statement was equally updated.

2. Methodological Clarity Regarding Treatment Impact:

Description: Adult pests were introduced only at the start, yet the discussion posits the material kills pests. Clarification is needed on how the results are interpreted concerning the pest's full life cycle (including eggs and larvae inside grains).

Proposed Solution: Adding clarification in the methodology or discussion stating that the design measures the cumulative ability of the treatment to suppress pest population development within the semi-closed environment, through its effect on adults and possibly by creating unfavorable conditions for larvae.

Response: Thank you for your remark, dear Editor. The clarification proposed was taken into consideration in the revised manuscript”.

3. Improving Results Presentation (Figures):

Description: Many figures (e.g., 2, 3, 4, 5) are overly complex with numerous plot lines, making it difficult for the reader to follow key outcomes.

Proposed Solution: It is suggested to split complex figures into simpler sub-figures (e.g., separating maize from beans, or presenting storage materials separately). Bar charts could help present comparisons more clearly at specific time points.

Response: We appreciate your remark dear Editor. We split complex figures into simpler sub-figures using bar charts to present storage materials separately.

4. Deepening the Discussion:

Description: The discussion could be enriched by a preliminary mention of the method’s potential economic feasibility compared to alternatives, and its expected performance under different humidity or temperature conditions than those in the lab.

Proposed Solution: Adding a short paragraph in the concluding section or discussion touching on these points would enhance the applied value of the research.

Response: We are so grateful for your remark, dear Editor.

Editorial Notes

Language: Some minor typographical and grammatical errors require correction (e.g., "obectus" should be corrected to "obtectus").

Response: Thank you for your remark dear Editor. The typographical and grammatical errors were corrected in the revised manuscript.

References: The formatting of some references is incomplete or inconsistent (e.g., references 1, 22, 39). They should be reviewed and standardized according to the journal’s style guide.

Response: Many thanks for your remark, dear Editor. The references 1, 22 and 39 were formatted and standardized according to the journal’s style guide.

Clarity: Ensure all abbreviations (GJ, PE, PP) are defined upon first use, and that all figure legends are complete and clear.

Response: We are so grateful for your remark, dear Editor.

---

## [Decision Letter · Decision Letter 2]

16 Feb 2026

Influence of biosilica treatments and storage receptacles on the quality of maize (Zea mays L.) and common bean (Phaseolus vulgaris L.) seeds during long-term storage

PONE-D-25-33584R2

Dear Dr. Zing Zing Bertrand,

We’re pleased to inform you that your manuscript has been judged scientifically suitable for publication and will be formally accepted for publication once it meets all outstanding technical requirements.

Kind regards,

Yêyinou Laura Estelle Loko

Academic Editor

PLOS One

Additional Editor Comments (optional):

Reviewers' comments:

Reviewer's Responses to Questions

**Comments to the Author**

Reviewer #4: All comments have been addressed

2. Is the manuscript technically sound, and do the data support the conclusions?

Reviewer #4: Yes

3. Has the statistical analysis been performed appropriately and rigorously?

Reviewer #4: Yes

4. Have the authors made all data underlying the findings in their manuscript fully available?

Reviewer #4: Yes

5. Is the manuscript presented in an intelligible fashion and written in standard English?

Reviewer #4: Yes

Reviewer #4: The researcher addressed all the provided feedback, and his contributions were valuable and insightful. Therefore, I recommend accepting the research for publication in the journal.

**Do you want your identity to be public for this peer review?** For information about this choice, including consent withdrawal, please see our Privacy Policy

Reviewer #4: **Yes:** Dr. Nabil Abo Kaf

---

## [Editor Report · Acceptance letter]

PONE-D-25-33584R2

PLOS One

Dear Dr. Zing Zing,

I'm pleased to inform you that your manuscript has been deemed suitable for publication in PLOS One. Congratulations! Your manuscript is now being handed over to our production team.

Kind regards,

on behalf of

Dr. Yêyinou Laura Estelle Loko

Academic Editor

PLOS One